# Improving Generalization and Stability of Generative Adversarial Networks

**Hoang Thanh-Tung**
hoangtha@deakin.edu.au

**Truyen Tran**
truyen.tran@deakin.edu.au

**Svetha Venkatesh**
svetha.venkatesh@deakin.edu.au

## Abstract

Generative Adversarial Networks (GANs) are one of the most popular tools for learning complex high dimensional distributions. However, generalization properties of GANs have not been well understood. In this paper, we analyze the generalization of GANs in practical settings. We show that discriminators trained on discrete datasets with the original GAN loss have poor generalization capability and do not approximate the theoretically optimal discriminator. We propose a zero-centered gradient penalty for improving the generalization of the discriminator by pushing it toward the optimal discriminator. The penalty guarantees the generalization and convergence of GANs. Experiments on synthetic and large scale datasets verify our theoretical analysis.

## 1 Introduction

GANs (Goodfellow et al., 2014) are one of the most popular tools for modeling high dimensional data. The original GAN is, however, highly unstable and often suffers from mode collapse. Much of recent researches has focused on improving the stability of GANs (Radford et al., 2015; Arjovsky et al., 2017; Heusel et al., 2017; Miyato et al., 2018; Karras et al., 2018). On the theoretical aspect, Nagarajan & Kolter (2017) proved that gradient based training of the original GAN is locally stable. Heusel et al. (2017) further proved that GANs trained with Two Timescale Update Rule (TTUR) converge to local equilibria. However, the generalization of GANs at local equilibria is not discussed in depth in these papers.

Arora et al. (2017) showed that the generator can win by remembering a polynomial number of training examples. The result implies that a low capacity discriminator cannot detect the lack of diversity. Therefore, it cannot teach the generator to approximate the target distribution. In section 4, we discuss the generalization capability of high capacity discriminators. We show that high capacity discriminators trained with the original GAN loss tends to overfit to the mislabeled samples in training dataset, guiding the generator toward collapsed equilibria (i.e. equilibria where the generator has mode collapse).

Arora et al. (2018) proposed to measure the generalization capability of GAN by estimating the number of modes in the model distribution using the birthday paradox. Experiments on several datasets showed that the number of modes in the model distribution is several times greater than the number of training examples. The author concluded that although GANs might not be able to learn distributions, they do exhibit some level of generalization. Our analysis shows that poor generalization comes from the mismatch between discriminators trained on discrete finite datasets and the theoretically optimal discriminator. We propose a zero-centered gradient penalty for improving the generalization capability of (high capacity) discriminators. Our zero-centered gradient penalty pushes the discriminator toward the optimal one, making GAN to converge to equilibrium with good generalization capability.

Our contributions are as follow:

1. We show that discriminators trained with the original GAN loss have poor generalization capability. Poor generalization in the discriminator prevents the generator from learning the target distribution.

2. We show that the original GAN objective encourages gradient exploding in the discriminator. Gradient exploding in the discriminator can lead to mode collapse in the generator.

3. We propose a zero-centered gradient penalty (0-GP) for improving the generalization capability of the discriminator. We show that non-zero centered GP and the zero-centered GP proposed in Mescheder et al. (2018) cannot make the discriminator generalize. Our 0-GP helps GANs to converge to generalizable equilibria. Theoretical results are verified on real world datasets.

4. We show that 0-GP helps the discriminator to distribute its capacity more equally between regions of the space, effectively preventing mode collapse. Experiments on synthetic and real world datasets verify that 0-GP can prevent mode collapse. GANs with 0-GP is much more robust to changes in hyper parameters, optimizers, and network architectures than the original GAN and GANs with other gradient penalties.

Table 1 compares the key properties of our 0-GP with one centered GP (1-GP) (Gulrajani et al., 2017) and zero centered GP on real/fake samples only (0-GP-sample) (Mescheder et al., 2018).

NOTATIONS

| | |
|---|---|
| $p_r$ | the target distribution |
| $p_g$ | the model distribution |
| $p_z$ | the noise distribution |
| $d_x$ | the dimensionality of a data sample (real or fake) |
| $d_z$ | the dimensionality of a noise sample |
| $supp(p)$ | the support of distribution $p$ |
| $\boldsymbol{x} \sim p_r$ | a real sample |
| $\boldsymbol{z} \sim p_z$ | a noise vector drawn from the noise distribution $p_z$ |
| $\boldsymbol{y} = G(\boldsymbol{z})$ | a generated sample |
| $\mathcal{D}_r = \{\boldsymbol{x}_1, ..., \boldsymbol{x}_n\}$ | the set of $n$ real samples |
| $\mathcal{D}_g^{(t)} = \left\{\boldsymbol{y}_1^{(t)}, ..., \boldsymbol{y}_m^{(t)}\right\}$ | the set of $m$ generated samples at step $t$ |
| $\mathcal{D}^{(t)} = \mathcal{D}_r \cup \mathcal{D}_g^{(t)}$ | the training dataset at step $t$ |

## 2 RELATED WORKS

Gradient penalties are widely used in GANs literature. There are a plethora of works on using gradient penalty to improve the stability of GANs (Mescheder et al., 2018; Gulrajani et al., 2017; Petzka et al., 2018; Roth et al., 2017; Qi, 2017). However, these works mostly focused on making the training of GANs stable and convergent. Our work aims to improve the generalization capability of GANs via gradient regularization.

Arora et al. (2018) showed that the number of modes in the model distribution grows linearly with the size of the discriminator. The result implies that higher capacity discriminators are needed for better approximation of the target distribution. Zhang et al. (2018) studied the tradeoff between generalization and discrimination in GANs. The authors showed that generalization is guaranteed if the discriminator set is small enough. In practice, rich discriminators are usually used for better discriminative power. Our GP makes rich discriminators generalizable while remaining discriminative.

Although less mode collapse is not exactly the same as generalization, the ability to produce more diverse samples implies better generalization. There are a large number of papers on preventing mode collapse in GANs. Radford et al. (2015); Salimans et al. (2016) introduced a number of empirical tricks to help stabilizing GANs. Arjovsky & Bottou (2017) showed the importance of divergences in GAN training, leading to the introduction of Wasserstein GAN (Arjovsky et al., 2017). The use of weak divergence is further explored by Mroueh & Sercu (2017); Mroueh et al. (2018). Lucas et al. (2018) advocated the use of mixed-batches, mini-batches of real and fake data,

| GP | Formula | Improve generalization | Prevent grad expoding | Convergence guarantee |
|---|---|---|---|---|
| Our 0-GP | $\lambda\mathbb{E}_{\boldsymbol{v}\in\mathcal{C}}[\|(\nabla D)_{\boldsymbol{v}}\|^2]$, $\mathcal{C}$ from $\boldsymbol{y}$ to $\boldsymbol{x}$ | ✓ | ✓ | ✓ |
| 1-GP | $\lambda\mathbb{E}_{\tilde{\boldsymbol{x}}}[(\|(\nabla D)_{\tilde{\boldsymbol{x}}}\| - 1)^2]$, where $\tilde{\boldsymbol{x}} = \alpha\boldsymbol{x} + (1-\alpha)\boldsymbol{y}$ | ✗ | ✓ | ✗ |
| 0-GP-sample | $\lambda\mathbb{E}_{\boldsymbol{v}\in\mathcal{D}}[\|(\nabla D)_{\boldsymbol{v}}\|^2]$ | ✗ | ✗ | ✓ |

Table 1: Summary of different gradient penalties

to smooth out the loss surface. The method exploits the distributional information in a mini-batch to prevent mode collapse. VEEGAN (Srivastava et al., 2017) uses an inverse of the generator to map the data to the prior distribution. The mismatch between the inverse mapping and the prior is used to detect mode collapse. If the generator can remember the entire training set, then the inverse mapping can be arbitrarily close the the prior distribution. It suggests that VEEGAN might not be able to help GAN to generalize outside of the training dataset. Our method helps GANs to discover unseen regions of the target distribution, significantly improve the diversity of generated samples.

## 3 BACKGROUND

In the original GAN, the discriminator $D$ maximizes the following objective

$$\mathcal{L} = \mathbb{E}_{\boldsymbol{x}\sim p_r}[\log(D(\boldsymbol{x}))] + \mathbb{E}_{\boldsymbol{z}\sim p_z}[\log(1 - D(G(\boldsymbol{z})))] \tag{1}$$

Goodfellow et al. (2014) showed that if the density functions $p_g$ and $p_r$ are known, then for a fixed generator $G$ the optimal discriminator is

$$D^*(\boldsymbol{v}) = \frac{p_r(\boldsymbol{v})}{p_r(\boldsymbol{v}) + p_g(\boldsymbol{v})}, \forall \boldsymbol{v} \in supp(p_r) \cup supp(p_g) \tag{2}$$

In the beginning of the training, $p_g$ is very different from $p_r$ so we have $p_r(\boldsymbol{x}) \gg p_g(\boldsymbol{x})$, for $\boldsymbol{x} \in \mathcal{D}_r$ and $p_g(\boldsymbol{y}) \gg p_r(\boldsymbol{y})$, for $\boldsymbol{y} \in \mathcal{D}_g$. Therefore, in the beginning of the training $D^*(\boldsymbol{x}) \approx 1$, for $\boldsymbol{x} \in \mathcal{D}_r$ and $D^*(\boldsymbol{y}) \approx 0$, for $\boldsymbol{y} \in \mathcal{D}_g$. As the training progresses, the generator will bring $p_g$ closer to $p_r$. The game reaches the global equilibrium when $p_r = p_g$. At the global equilibrium, $D^*(\boldsymbol{v}) = \frac{1}{2}, \forall \boldsymbol{v} \in supp(p_r) \cup supp(p_g)$. One important result of the original paper is that, if the discriminator is optimal at every step of the GAN algorithm, then $p_g$ converges to $p_r$.

In practice, density functions are not known and the optimal discriminator is approximated by optimizing the classification performance of a parametric discriminator $D(\cdot; \boldsymbol{\theta}_D)$ on a discrete finite dataset $\mathcal{D} = \mathcal{D}_r \cup \mathcal{D}_g$. We call a discriminator trained on a discrete finite dataset an empirical discriminator. The empirically optimal discriminator is denoted by $\hat{D}^*$.

Arora et al. (2017) defined generalization of a divergence $d$ as follow: A divergence $d$ is said to have generalization error $\epsilon$ if

$$|d(\mathcal{D}_g, \mathcal{D}_r) - d(p_g, p_r)| \le \epsilon \tag{3}$$

A discriminator $D$ defines a divergence between two distributions. The performance of a discriminator with good generalization capability on the training dataset should be similar to that on the entire data space. In practice, generalization capability of $D$ can be estimated by measuring the difference between its performance on the training dataset and a held-out dataset.

## 4 GENERALIZATION CAPABILITY OF DISCRIMINATORS

### 4.1 THE EMPIRICALLY OPTIMAL DISCRIMINATOR DOES NOT APPROXIMATE THE THEORETICALLY OPTIMAL DISCRIMINATOR

It has been observed that if the discriminator is too good at discriminating real and fake samples, the generator cannot learn effectively (Goodfellow et al., 2014; Arjovsky & Bottou, 2017). The phenomenon suggests that $\hat{D}^*$ does not well approximate $D^*$, and does not guarantee the convergence of $p_g$ to $p_r$. In the following, we clarify the mismatch between $\hat{D}^*$ and $D^*$, and its implications.

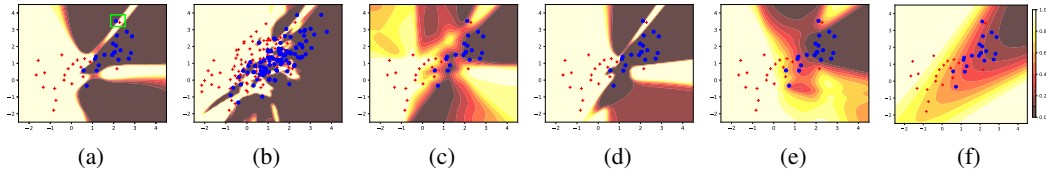

Figure 1: Value surfaces of discriminators trained for 10,000 iterations with different gradient penalties, on samples from two Gaussian distributions. The discriminator is a 2 hidden layer MLP with 64 hidden neurons.(a) No GP. (b) No GP with more samples. (c) One-centered GP (1-GP) with $\lambda = 1$. (d) Zero-centered GP on real/fake samples only (0-GP-sample) with $\lambda = 1$. (e) Our zero-centered GP with $\lambda = 1$. (f) Theoretically optimal discriminator computed using Eqn. 2.

**Proposition 1.** *The two datasets $\mathcal{D}_r$ and $\mathcal{D}_g^{(t)}$ are disjoint with probability $1$ regardless of how close the two distributions $p_r$ and $p_g^{(t)}$ are.*

*Proof.* See appendix A. $\qquad\qquad\qquad\qquad\qquad\qquad\qquad\qquad\qquad\qquad\qquad\qquad\qquad\qquad$ $\square$

$\mathcal{D}_r$ and $\mathcal{D}_g^{(t)}$ are disjoint with probability 1 even when $p_g$ and $p_r$ are exactly the same. $\hat{D}^*$ perfectly classifies the real and the fake datasets, and $\hat{D}^*(\boldsymbol{x}) = 1, \forall \boldsymbol{x} \in \mathcal{D}_r$ , $\hat{D}^*(\boldsymbol{y}) = 0, \forall \boldsymbol{y} \in \mathcal{D}_g^{(t)}$. The value of $\hat{D}^*$ on $\mathcal{D}^{(t)}$ does not depend on the distance between the two distributions and does not reflect the learning progress. The value of $\hat{D}^*$ on the training dataset approximates that of $D^*$ in the beginning of the learning process but not when the two distributions are close. When trained using gradient descent on a discrete finite dataset with the loss in Eqn. 1, the discriminator $D$ is pushed toward $\hat{D}^*$, not $D^*$. This behavior does not depend on the size of training set (see Fig. 1a, 1b), implying that *the original GAN is not guaranteed to converge to the target distribution even when given enough data.*

### 4.2    EMPIRICAL DISCRIMINATORS HAVE POOR GENERALIZATION CAPABILITY

When the generator gets better, generated samples are more similar to samples from the target distribution. However, regardless of their quality, generated samples are still labeled as fake in Eqn. 1. The training dataset $\mathcal{D}$ is a bad dataset as it contains many mislabeled examples. A discriminator trained on such dataset will overfit to the mislabeled examples and has poor generalization capability. It will misclassify unseen samples and cannot teach the generator to generate these samples.

Figure 1a and 1b demonstrate the problem on a synthetic dataset consisting of samples from two Gaussian distributions. The discriminator in Fig. 1a overfits to the small dataset and does not generalize to new samples in Fig. 1b. Although the discriminator in Fig. 1b was trained on a larger dataset which is sufficient to characterize the two distributions, it still overfits to the data and its value surface is very different from that of the theoretically optimal discriminator in Fig. 1f.

An overfitted discriminator does not guide the model distribution toward target distribution but toward the real samples in the dataset. This explains why the original GAN usually exhibits mode collapse behavior. Finding the empirically optimal discriminator using gradient descent usually requires many iterations. Heuristically, overfitting can be alleviated by limiting the number of discriminator updates per generator update. Goodfellow et al. (2014) recommended to update the discriminator once every generator update. In the next subsection, we show that limiting the number of discriminator updates per generator update prevents the discriminator from overfitting.

### 4.2.1    $\epsilon$-OPTIMAL DISCRIMINATORS

$\hat{D}^*$ is costly to find and maintain. We consider here a weaker notion of optimality which can be achieved in practical settings.

**Definition 1** ($\epsilon$-optimal discriminator). *Given two disjoint datasets $\mathcal{D}_r$ and $\mathcal{D}_g$, and a number $\epsilon > 0$, a discriminator $D$ is $\epsilon$-optimal if*

$$
\begin{aligned}
D(\boldsymbol{x}) &\geq \frac{1}{2} + \frac{\epsilon}{2}, \forall \boldsymbol{x} \in \mathcal{D}_r \\
D(\boldsymbol{y}) &\leq \frac{1}{2} - \frac{\epsilon}{2}, \forall \boldsymbol{y} \in \mathcal{D}_g
\end{aligned}
$$

As observed in Goodfellow et al. (2014), $\hat{D}^*$ does not generate usable gradient for the generator. Goodfellow et al. proposed the non-saturating loss for the generator to circumvent this vanishing gradient problem. For an $\epsilon$-optimal discriminator, if $\epsilon$ is relatively small, then the gradient of the discriminator w.r.t. fake datapoints might not vanish and can be used to guide the model distribution toward the target distribution.

**Proposition 2.** *Given two disjoint datasets $\mathcal{D}_r$ and $\mathcal{D}_g$, and a number $\epsilon > 0$, an $\epsilon$-optimal discriminator $D_\epsilon$ exists and can be constructed as a one hidden layer MLP with $\mathcal{O}(d_x(m+n))$ parameters.*

*Proof.* See appendix B. $\qquad\qquad\qquad\qquad\qquad\qquad\qquad\qquad\qquad\qquad\qquad\qquad\square$

Because deep networks are more powerful than shallow ones, the size of a deep $\epsilon$-optimal discriminator can be much smaller than $\mathcal{O}(d_x(m+n))$. From the formula, the size of a shallow $\epsilon$-optimal discriminator for real world datasets ranges from a few to hundreds of millions parameters. That is comparable to the size of discriminators used in practice. Arjovsky & Bottou (2017) showed that even when the generator can generate realistic samples, a discriminator that can perfectly classify real and fake samples can be found easily using gradient descent. The experiment verified that $\epsilon$-optimal discriminator can be found using gradient descent in practical settings.

We observe that the norm of the gradient w.r.t. the discriminator's parameters decreases as fakes samples approach real samples. If the discriminator's learning rate is fixed, then the number of gradient descent steps that the discriminator has to take to reach $\epsilon$-optimal state should increase.

**Proposition 3.** *Alternating gradient descent with the same learning rate for discriminator and generator, and fixed number of discriminator updates per generator update (Fixed-Alt-GD) cannot maintain the (empirical) optimality of the discriminator.*

Fixed-Alt-GD decreases the discriminative power of the discriminator to improve its generalization capability. The proof for linear case is given in appendix C.

In GANs trained with Two Timescale Update Rule (TTUR) (Heusel et al., 2017), the ratio between the learning rate of the discriminator and that of the generator goes to infinity as the iteration number goes to infinity. Therefore, the discriminator can learn much faster than the generator and might be able to maintain its optimality throughout the learning process.

### 4.2.2 GRADIENT EXPLODING IN $\epsilon$-OPTIMAL DISCRIMINATORS

Let's consider a simplified scenario where the real and the fake datasets each contains a single datapoint: $\mathcal{D}_r = \{\boldsymbol{x}\}$, $\mathcal{D}_g^{(t)} = \{\boldsymbol{y}^{(t)}\}$. Updating the generator according to the gradient from the discriminator will push $\boldsymbol{y}^{(t)}$ toward $\boldsymbol{x}$. The absolute value of directional derivative of $D$ in the direction $\boldsymbol{u} = \boldsymbol{x} - \boldsymbol{y}^{(t)}$, at $\boldsymbol{x}$ is

$$
|(\nabla_{\boldsymbol{u}} D)_{\boldsymbol{x}}| = \lim_{\boldsymbol{y}^{(t)} \xrightarrow{\boldsymbol{u}} \boldsymbol{x}} \frac{\left| D(\boldsymbol{x}) - D(\boldsymbol{y}^{(t)}) \right|}{\left\| \boldsymbol{x} - \boldsymbol{y}^{(t)} \right\|}
$$

If $D$ is always $\epsilon$-optimal, then $\left| D(\boldsymbol{x}) - D(\boldsymbol{y}^{(t)}) \right| \geq \epsilon, \forall t \in \mathbb{N}$, and

$$
|(\nabla_{\boldsymbol{u}} D)_{\boldsymbol{x}}| \geq \lim_{\boldsymbol{y}^{(t)} \xrightarrow{\boldsymbol{u}} \boldsymbol{x}} \frac{\epsilon}{\left\| \boldsymbol{x} - \boldsymbol{y}^{(t)} \right\|} = \infty
$$

The directional derivate of the $\epsilon$-optimal discriminator explodes as the fake datapoint approaches the real datapoint. Directional derivative exploding implies gradient exploding at datapoints on the line

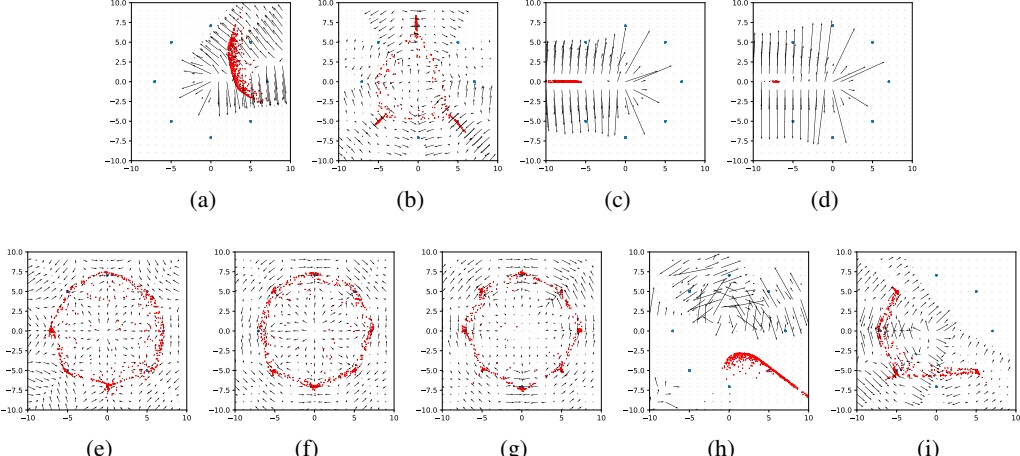

Figure 2: Gradient w.r.t. the input of the discriminator of a GAN trained with different gradient penalties. The vector associated with a datapoint $v$ points in the direction that increases the value of $\log(D(v))$ the fastest. The discriminator is a 2 hidden layer MLP with 512 hidden neurons. The discriminator is updated once every generator update. SGD is used for optimization. (a), (b) No GP, iter. 1000 and 10,000. (c), (d) No GP with TTUR, iter. 1,000 and 10,000. (e) Our 0-GP with $\lambda = 10$, iter. 10,000. (f), (g) Our 0-GP with TTUR and $\lambda = 10$, iter. 10,000 and 20,000. (h) 1-GP with $\lambda = 10$, iter. 10,000. (i) 0-GP-sample with $\lambda = 10$, iter. 10,000.

segment connecting $x$ and $y^{(t)}$. If in the next iteration, the generator produces a sample in a region where the gradient explodes, then the gradient w.r.t. the generator's parameters explodes.

Let's consider the following line integral

$$\int_{\mathcal{C}} (\nabla D)_v \cdot ds = D(x) - D(y^{(t)}) \tag{4}$$

where $\mathcal{C}$ is the line segment from $y^{(t)}$ to $x$. As the model distribution gets closer to the target distribution, the length of $\mathcal{C}$ should be non increasing. Therefore, maximizing $D(x) - D(y^{(t)})$, or the discriminative power of $D$, leads to the maximization of the directional derivative of $D$ in the direction $ds$. The original GAN loss makes $D$ to maximize its discriminative power, encouraging gradient exploding to occur.

Gradient exploding happens in the discriminator trained with TTUR in Fig. 2c and 2d. Because TTUR can help the discriminator to maintain its optimality, gradient exploding happens and persists throughout the training process. Without TTUR, the discriminator cannot maintain its optimality so gradient exploding can happen sometimes during the training but does not persist (Fig. 2a and 2b). Because of the saturated regions in the sigmoid function used in neural network based discriminators, the gradient w.r.t. datapoints in the training set could vanishes. However, gradient exploding must happen at some datapoints on the path between a pair of samples, where the sigmoid function does not saturate. In Fig. 1a, gradient exploding happens near the decision boundary.

In practice, $\mathcal{D}_r$ and $\mathcal{D}_g$ contain many datapoints and the generator is updated using the average of gradients of the discriminator w.r.t. fake datapoints in the mini-batch. If a fake datapoint $y_0$ is very close to a real datapoint $x_0$, the gradient $(\nabla D)_{y_0}$ might explode. When the average gradient is computed over the mini-batch, $(\nabla D)_{y_0}$ outweighs other gradients. The generator updated with this average gradient will move many fake datapoints in the direction of $(\nabla D)_{y_0}$, toward $x_0$, making mode collapse visible.

## 5 IMPROVING GENERALIZATION CAPABILITY OF EMPIRICAL DISCRIMINATORS

Although the theoretically optimal discriminator $D^*$ is generalizable, the original GAN loss does not push empirical discriminators toward $D^*$. We aim to improve the generalization capability of empirical discriminators by pushing them toward $D^*$.

### 5.1 PUSHING EMPIRICAL DISCRIMINATORS TOWARD $D^*$

For any input $\boldsymbol{v} \in supp(p_r) \cup supp(p_g)$, the value of $D^*(\boldsymbol{v})$ goes to $\frac{1}{2}$ and the gradient $(\nabla D)_{\boldsymbol{v}}$ goes to $\mathbf{0}$ as $p_g$ approaches $p_r$. Consider again the line integral in Eqn. 4. As $D^*(\boldsymbol{x})$ and $D^*(\boldsymbol{y})$ approach $\frac{1}{2}$ for all $\boldsymbol{x} \in supp(p_r)$ and $\boldsymbol{y} \in supp(p_g)$, we have

$$D^*(\boldsymbol{x}) - D^*(\boldsymbol{y}) = \int_{\mathcal{C}} (\nabla D^*)_{\boldsymbol{v}} \cdot d\boldsymbol{s} \to 0 \tag{5}$$

for all pairs of $\boldsymbol{x}$ and $\boldsymbol{y}$ and all paths $\mathcal{C}$ from $\boldsymbol{y}$ to $\boldsymbol{x}$. That means, the discriminative power of $D^*$ must decrease as the two distributions become more similar.

To push an empirical discriminator $D$ toward $D^*$, we force $D$ to satisfy two requirements:

1. $(\nabla D)_{\boldsymbol{v}} \to \mathbf{0}, \forall\, \boldsymbol{v} \in supp(p_r) \cup supp(p_g)$
2. $D(\boldsymbol{x}) - D(\boldsymbol{y}) = \int_{\mathcal{C}} (\nabla D)_{\boldsymbol{v}} \cdot d\boldsymbol{s} \to 0, \forall\, \boldsymbol{x} \sim p_r, \boldsymbol{y} \sim p_g, \mathcal{C}$ from $\boldsymbol{y}$ to $\boldsymbol{x}$

### 5.2 ZERO-CENTERED GRADIENT PENALTY

The first requirement can be implemented by sampling some datapoints $\boldsymbol{v} \in supp(p_r) \cup supp(p_g)$ and force $(\nabla D)_{\boldsymbol{v}}$ to be $\mathbf{0}$. The second requirement can be implemented by sampling pairs of real and fake datapoints $(\boldsymbol{x}, \boldsymbol{y})$ and force $D(\boldsymbol{x}) - D(\boldsymbol{y})$ to be 0. The two requirements can be added to the discriminator's objective as follows

$$\hat{\mathcal{L}} = \mathcal{L} - \lambda_1 \mathbb{E}_{\boldsymbol{v}}[\|(\nabla D)_{\boldsymbol{v}}\|^2] - \lambda_2 \mathbb{E}_{\boldsymbol{x},\boldsymbol{y}}[(D(\boldsymbol{x}) - D(\boldsymbol{y}))^2]$$

where $\mathcal{L}$ is the objective in Eqn. 1. However, as discussed in section 4.2.2, an $\epsilon$-optimal discriminator can have zero gradient on the training dataset and have gradient exploding outside of the training dataset. The gradient norm could go to infinity even when $D(\boldsymbol{x}) - D(\boldsymbol{y})$ is small. Regulating the difference between $D(\boldsymbol{x})$ and $D(\boldsymbol{y})$ is not an efficient way to prevent gradient exploding.

We want to prevent gradient exploding on every path in $supp(p_r) \cup supp(p_g)$. Because $(\nabla D^*)_{\boldsymbol{v}} \to \mathbf{0}$ for all $\boldsymbol{v} \in supp(p_r) \cup supp(p_g)$ as $p_g$ approach $p_r$, we could push the gradient w.r.t. every datapoint on every path $\mathcal{C} \in supp(p_r) \cup supp(p_g)$ toward $\mathbf{0}$. We note that, if $(\nabla D)_{\boldsymbol{v}} \to \mathbf{0}, \forall\, \boldsymbol{v} \in \mathcal{C}$ then $\int_{\mathcal{C}} (\nabla D)_{\boldsymbol{v}} \cdot d\boldsymbol{s} \to 0$. Therefore, the two requirements can be enforced by a single zero-centered gradient penalty of the form

$$\lambda \mathbb{E}_{\boldsymbol{v} \in \mathcal{C}}[\|(\nabla D)_{\boldsymbol{v}}\|^2]$$

The remaining problem is how to find the path $\mathcal{C}$ from a fake to a real sample which lies inside $supp(p_r) \cup supp(p_g)$. Because we do not have access to the full supports of $p_r$ and $p_g$, and the supports of two distributions could be disjoint in the beginning of the training process, finding a path which lies completely inside the support is infeasible.

In the current implementation, we approximate $\mathcal{C}$ with the straight line connecting a pair of samples, although there is no guarantee that all datapoints on that straight line are in $supp(p_r) \cup supp(p_g)$. That results in the following objective

$$\mathcal{L}_{0-GP} = \mathcal{L} - \lambda \mathbb{E}_{\tilde{\boldsymbol{x}}}[\|(\nabla D)_{\tilde{\boldsymbol{x}}}\|^2] \tag{6}$$

where $\tilde{\boldsymbol{x}} = \alpha\boldsymbol{x} + (1 - \alpha)\boldsymbol{y}$, $\boldsymbol{x} \sim p_r$, $\boldsymbol{y} \sim p_g$, and $\alpha \sim \mathcal{U}(0, 1)$ [1]. We describe a more sophisticated way of finding a better path in appendix F.

---

[1] Wu et al. (2018) independently proposed the Wasserstein divergence for WGAN which uses a gradient penalty of similar form. Although the two penalties have similar approximate form, they have different motivations and addresses different problems in GANs.

The larger $\lambda$ is, the stronger $(\nabla D)_{\tilde{x}}$ is pushed toward $\mathbf{0}$. If $\lambda$ is 0, then the discriminator will only focus on maximizing its discriminative power. If $\lambda$ approaches infinity, then the discriminator has maximum generalization capability and no discriminative power. $\lambda$ controls the tradeoff between discrimination and generalization in the discriminator.

## 5.3 GENERALIZATION CAPABILITY OF DIFFERENT GRADIENT PENALTIES

Mescheder et al. (2018) proposed to force the gradient w.r.t. datapoints in the real and/or fake dataset(s) to be $\mathbf{0}$ to make the training of GANs convergent. In section 4, we showed that for discrete training dataset, an empirically optimal discriminator $\hat{D}^*$ always exists and could be found by gradient descent. Although $(\nabla \hat{D}^*)_v = \mathbf{0}, \forall\ v \in \mathcal{D}$, $\hat{D}^*$ does not satisfy the requirement in Eqn. 5 and have gradient exploding when some fake datapoints approach a real datapoint. The discriminators in Fig. 1a, 1b, 1d, 2c and 2d have vanishingly small gradients on datapoints in the training dataset and very large gradients outside. They have poor generalization capability and cannot teach the generator to generate unseen real datapoints. Therefore, *zero-centered gradient penalty on samples from $p_r$ and $p_g$ only cannot help improving the generalization of the discriminator.*

*Non-zero centered GPs do not push an empirical discriminator toward $D^*$ because the gradient does not converge to $\mathbf{0}$.* A commonly used non-zero centered GP is the one-centered GP (1-GP) (Gulrajani et al., 2017) which has the following form

$$\lambda \mathbb{E}_{\tilde{x}}[(\|(\nabla D)_{\tilde{x}}\| - 1)^2] \tag{7}$$

where $\tilde{x} = \alpha x + (1 - \alpha)y$, $x \sim p_r$, $y \sim p_g$, and $\alpha \sim \mathcal{U}(0, 1)$. Although the initial goal of 1-GP was to enforce Lipschitz constraint on the discriminator [2], Fedus et al. (2018) found that 1-GP prevents gradient exploding, making the original GAN more stable. 1-GP forces the norm of gradients w.r.t. datapoints on the line segment connecting $x$ and $y$ to be 1. If all gradients on the line segment have norm 1, then the line integral in Eqn. 4 could be as large as $\|x - y\|$. Because the distance between random samples grows with the dimensionality, in high dimensional space $\|x - y\|$ is greater than 1 with high probability. The discriminator could maximize the value of the line integral without violating the Lipschitz constraint. The discriminator trained with 1-GP, therefore, can overfit to the training data and have poor generalization capability.

## 5.4 CONVERGENCE ANALYSIS FOR ZERO-CENTERED GRADIENT PENALTY

Mescheder et al. (2018) showed that zero-centered GP on real and/or fake samples (0-GP-sample) makes GANs convergent. The penalty is based on the convergence analysis for the Dirac GAN, an 1-dimensional linear GAN which learns the Dirac distribution. The intuition is that when $p_g$ is the same as $p_r$, the gradient of the discriminator w.r.t. the fake datapoints (which are also real datapoints) should be $\mathbf{0}$ so that generator will not move away when being updated using this gradient. If the gradient from the discriminator is not $\mathbf{0}$, then the generator will oscillate around the equilibrium.

Our GP forces the gradient w.r.t. all datapoints on the line segment between a pair of samples (including the two endpoints) to be $\mathbf{0}$. As a result, our GP also prevents the generator from oscillating. Therefore, our GP has the same convergence guarantee as the 0-GP-sample.

## 5.5 ZERO-CENTERED GRADIENT PENALTY IMPROVES CAPACITY DISTRIBUTION

Discriminators trained with the original GAN loss tends to focus on the region of the where fake samples are close to real samples, ignoring other regions. The phenomenon can be seen in Fig. 2a, 2b, 2c, 2d, 2h and 2i. Gradients in the region where fake samples are concentrated are large while gradients in other regions, including regions where real samples are located, are very small. The generator cannot discover and generate real datapoints in regions where the gradient vanishes.

When trained with the objective in Eqn. 6, the discriminator will have to balance between maximizing $\mathcal{L}$ and minimizing the GP. For finite $\lambda$, the GP term will not be exactly 0. Let $\gamma = \lambda \mathbb{E}_{\tilde{x}}[\|(\nabla D)_{\tilde{x}}\|^2]$. Among discriminators with the same value of $\gamma$, gradient descent will find the discriminator that maximizes $\mathcal{L}$. As discussed in section 4.2.2, maximizing $\mathcal{L}$ leads to the

---

[2]Petzka et al. (2018) pointed out that 1-GP is based on the wrong intuition that the gradient of the optimal critic must be 1 everywhere under $p_r$ and $p_g$. The corrected GP is based on the definition of Lipschitzness.

maximization of norms of gradients on the path from $\boldsymbol{y}$ to $\boldsymbol{x}$. The discriminator should maximize the value $\eta = \mathbb{E}_{\tilde{\boldsymbol{x}}}[\|(\nabla D)_{\tilde{\boldsymbol{x}}}\|]$. If $\gamma$ is fixed then $\eta$ is maximized when $\|\nabla D_{\tilde{\boldsymbol{x}}^{(i)}}\| = \|\nabla D_{\tilde{\boldsymbol{x}}^{(j)}}\|, \forall\, i, j$ (Cauchy-Schwarz inequality). Therefore, our zero-centered GP encourages the gradients at different regions of the real data space to have the same norm. The capacity of $D$ is distributed more equally between regions of the real data space, effectively reduce mode collapse. The effect can be seen in Fig. 2e and 2f.

1-GP encourages $|\|\nabla D_{\tilde{\boldsymbol{x}}^{(i)}}\| - 1| = |\|\nabla D_{\tilde{\boldsymbol{x}}^{(j)}}\| - 1|, \forall\, i, j$. That allows gradient norms to be smaller than 1 in some regions and larger than 1 in some other regions. The problem can be seen in Fig. 2h.

## 6 EXPERIMENTS

The code is made available at https://github.com/htt210/GeneralizationAndStabilityInGANs.

### 6.1 ZERO-CENTERED GRADIENT PENALTY PREVENTS OVERFITTING

To test the effectiveness of gradient penalties in preventing overfitting, we designed a dataset with real and fake samples coming from two Gaussian distributions and trained a MLP based discriminator on that dataset. The result is shown in Fig. 1. As predicted in section 5.3, 0-GP-sample does not help to improve generalization. 1-GP helps to improve generalization. The value surface in Fig. 1c is smoother than that in Fig. 1a. However, as discussed in section 5.3, 1-GP cannot help much in higher dimensional space where the pair-wise distances are large. The discriminator trained with our 0-GP has the best generalization capability, with a value surface which is the most similar to that of the theoretically optimal one.

We increased the number of discriminator updates per generator update to 5 to see the effect of GPs in preventing overfitting. On the MNIST dataset, GAN without GP and with other GPs cannot learn anything after 10,000 iterations. GAN with our 0-GP can still learn normally and start produce recognizable digits after only 1,000 iterations. The result confirms that our GP is effective in preventing overfitting in the discriminator.

### 6.2 ZERO-CENTERED GRADIENT PENALTY IMPROVES GENERALIZATION AND ROBUSTNESS OF GANS

#### SYNTHETIC DATA

We tested different gradient penalties on a number of synthetic datasets to compare their effectiveness. The first dataset is a mixture of 8 Gaussians. The dataset is scaled up by a factor of 10 to simulate the situation in high dimensional space where random samples are far from each other. The result is shown in Fig. 2. GANs with other gradient penalties all fail to learn the distribution and exhibit mode collapse problem to different extents. GAN with our 0-GP (GAN-0-GP) can successfully learn the distribution. Furthermore, GAN-0-GP can generate datapoints on the circle, demonstrating good generalization capability. The original GAN collapses to some disconnected modes and cannot perform smooth interpolation between modes: small change in the input result in large, unpredictable change in the output. GAN with zero-centered GP on real/fake samples only also exhibits the same "mode jumping" behavior. The behavior suggests that these GANs tend to remember the training dataset and have poor generalization capability. Fig. 9 in appendix D demonstrates the problem on MNIST dataset.

We observe that GAN-0-GP behaves similar to Wasserstein GAN as it first learns the overall structure of the distribution and then focuses on the modes. An evolution sequence of GAN-0-GP is shown in Fig. 5 in appendix D. Results on other synthetic datasets are shown in appendix D.

#### MNIST DATASET

The result on MNIST dataset is shown in Fig. 3. After 1,000 iterations, all other GANs exhibit mode collapse or cannot learn anything. GAN-0-GP is robust to changes in hyper parameters such

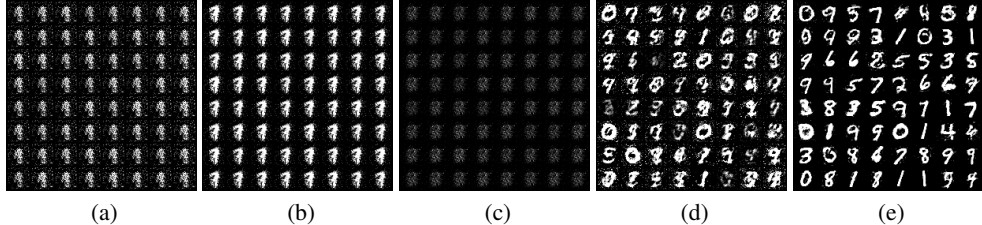

(a) (b) (c) (d) (e)

Figure 3: Result on MNIST. The networks have the same architectures with networks used in synthetic experiment. Batch normalization (Ioffe & Szegedy, 2015) was not used. Adam optimizer (Kingma & Ba, 2014) with $\beta_1 = 0.5, \beta_2 = 0.9$ was used. (a) No GP, iter. 1,000. (b) 0-GP-sample, $\lambda = 100$, iter. 1,000. (c) 1-GP, $\lambda = 100$, iter. 1,000. (d), (e) 0-GP, $\lambda = 100$, iter. 1,000 and 10,000.

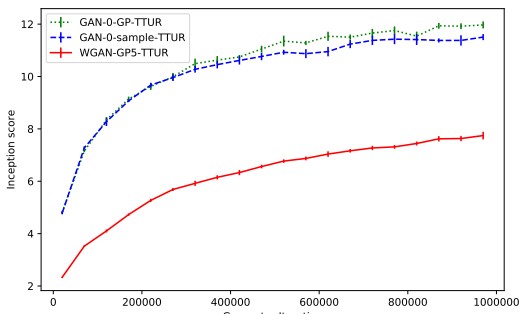

Figure 4: Inception score (Salimans et al., 2016) on ImageNet of GAN-0-GP, GAN-0-GP-sample, and WGAN-GP. The code for this experiment is adapted from Mescheder et al. (2018). We used $\lambda = 10$ for all GANs as recommended by Mescheder et al. The critic in WGAN-GP was updated 5 times per generator update. To improve convergence, we used TTUR with learning rates of 0.0001 and 0.0003 for the generator and discriminator, respectively.

as learning rate and optimizers. When Adam is initialized with large $\beta_1$, e.g. 0.9, GANs with other GPs cannot learn anything after many iterations. More samples are given in appendix D.

We observe that higher value of $\lambda$ improves the diversity of generated samples. For $\lambda = 50$, we observe some similar looking samples in the generated data. This is consistent with our conjecture that larger $\lambda$ leads to better generalization.

IMAGENET

When trained on ImangeNet (Deng et al., 2009), GAN-0-GP can produce high quality samples from all 1,000 classes. We compared our method with GAN with 0-GP-sample and WGAN-GP. GAN-0-GP-sample is able to produce samples of state of the art quality without using progressive growing trick (Karras et al., 2018). The result in Fig. 4 shows that our method consistently outperforms GAN-0-GP-sample. GAN-0-GP and GAN-0-GP-sample outperform WGAN-GP by a large margin. Image samples are given in appendix D.

## 7 CONCLUSION

In this paper, we clarify the reason behind the poor generalization capability of GAN. We show that the original GAN loss does not guide the discriminator and the generator toward a generalizable equilibrium. We propose a zero-centered gradient penalty which pushes empirical discriminators toward the optimal discriminator with good generalization capability. Our gradient penalty provides better generalization and convergence guarantee than other gradient penalties. Experiments on diverse datasets verify that our method significantly improves the generalization and stability of GANs.

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

## A    PROOF FOR PROPOSITION 1

For continuous random variable $V$, $\mathbb{P}(V = v) = 0$ for any $v$. The probability of finding a noise vector $z$ such that $G(z)$ is exactly equal to a real datapoint $x \in \mathcal{D}_r$ via random sampling is 0. Therefore, the probability of a real datapoint $x_i$ being in the fake dataset $\mathcal{D}_g$ is 0. Similarly, the probability of any fake datapoint being in the real dataset is 0.

$$\mathbb{P}(x \in \mathcal{D}_g^{(t)}) = 0, \forall x \in \mathcal{D}_r, t \in \mathbb{N}$$
$$\mathbb{P}(y \in \mathcal{D}_r) = 0, \forall y \in \mathcal{D}_g^{(t)}, t \in \mathbb{N}$$
$$\mathbb{P}(\mathcal{D}_r \cap \mathcal{D}_g^{(t)} = \emptyset) = 1, \forall t \in \mathbb{N} \tag{8}$$

Furthermore, due to the curse of dimensionality, the probability of sampling a datapoint which is close to another datapoint in high dimensional space also decrease exponentially. The distances between datapoints are larger in higher dimensional space. That suggests that it is easier to separate $\mathcal{D}_r$ and $\mathcal{D}_g^{(t)}$ in higher dimensional space.

## B CONSTRUCTING $\epsilon$-OPTIMAL DISCRIMINATORS

To make the construction process simpler, let's assume that samples are normalized:

$$\|\boldsymbol{x}_i\| = \|\boldsymbol{y}_j\| = 1, \forall \boldsymbol{x}_i \in \mathcal{D}_r, \boldsymbol{y}_j \in \mathcal{D}_g$$

Let's use the following new notations for real and fake samples:

$$
\begin{aligned}
\mathcal{D} &= \mathcal{D}_r \cup \mathcal{D}_g = \{\boldsymbol{v}_1, ..., \boldsymbol{v}_{m+n}\} \\
\boldsymbol{v}_i &= \begin{cases} \boldsymbol{x}_i, \text{ for } i = 1, ..., n \\ \boldsymbol{y}_{i-n}, \text{ for } i = n+1, ..., n+m \end{cases}
\end{aligned}
$$

We construct the $\epsilon$-optimal discriminator $D$ as a MLP with 1 hidden layer. Let $\boldsymbol{W}_1 \in \mathbb{R}^{(m+n) \times d_x}$ and $\boldsymbol{W}_2 \in \mathbb{R}^{m+n}$ be the weight matrices of $D$. The total number of parameters in $D$ is $d_x(m+n) + (m+n) = \mathcal{O}(d_x(m+n))$. We set the value of $\boldsymbol{W}_1$ as

$$
\boldsymbol{W}_1 = k \begin{bmatrix} \boldsymbol{v}_1^\top \\ \vdots \\ \boldsymbol{v}_{n+m}^\top \end{bmatrix}
$$

and $\boldsymbol{W}_2$ as

$$
\begin{aligned}
W_{2,i} &= \frac{1}{2} + \frac{\epsilon}{2} + \alpha, \text{ for } i = 1, ..., n \\
W_{2,i} &= \frac{1}{2} - \frac{\epsilon}{2} - \alpha, \text{ for } i = n+1, ..., n+m \\
\alpha &> 0
\end{aligned}
$$

Given an input $\boldsymbol{v} \in \mathcal{D}$, the output is computed as:

$$D(\boldsymbol{v}) = \boldsymbol{W}_2^\top \sigma(\boldsymbol{W}_1 \boldsymbol{v})$$

where $\sigma$ is the softmax function. Let $\boldsymbol{a} = \boldsymbol{W}_1 \boldsymbol{v}$, we have

$$
a_i = k \boldsymbol{v}_i^\top \boldsymbol{v} = \begin{cases} k, & \text{if } \boldsymbol{v} = \boldsymbol{v}_i \\ < k, & \text{if } \boldsymbol{v} \neq \boldsymbol{v}_i \end{cases}
$$

As $k \to \infty$, $\sigma(\boldsymbol{W}_1 \boldsymbol{v}_i)$ becomes a one-hot vector with the $i$-th element being 1, all other elements being 0. Thus, for large enough $k$, for any $\boldsymbol{v}_j \in \mathcal{D}$, the output of the network is

$$
D(\boldsymbol{v}_j) = \boldsymbol{W}_2^\top \sigma(\boldsymbol{W}_1 \boldsymbol{v}_j) \approx W_{2,j} = \begin{cases} > \frac{1}{2} + \frac{\epsilon}{2}, \text{ for } j = 1, ..., n \\ < \frac{1}{2} - \frac{\epsilon}{2}, \text{ for } j = n+1, ..., n+m \end{cases}
$$

$D$ is a $\epsilon$-optimal discriminator for dataset $\mathcal{D}$.

## C FIXED-ALT-GD CANNOT MAINTAIN THE OPTIMALITY OF $\epsilon$-DISCRIMINATORS

Let's consider the case where the real and the fake dataset each contain a single datapoint $\mathcal{D}_r = \{\boldsymbol{x}\}$, $\mathcal{D}_g^{(t)} = \boldsymbol{y}^{(t)}$, and the discriminator and the generator are linear:

$$
\begin{aligned}
D(\boldsymbol{v}) &= \boldsymbol{\theta}_D^\top \boldsymbol{v} \\
G(\boldsymbol{z}) &= \boldsymbol{\theta}_G \boldsymbol{z}
\end{aligned}
$$

and the objective is also linear (Wasserstein GAN's objective):

$$\mathcal{L}_{\mathcal{W}} = \mathbb{E}_{\boldsymbol{x} \in \mathcal{D}_r}[D(\boldsymbol{x})] - \mathbb{E}_{\boldsymbol{y} \in \mathcal{D}_g^{(t)}}[D(\boldsymbol{y})]$$
$$= D(\boldsymbol{x}) - D(\boldsymbol{y}^{(t)})$$

The same learning rate $\alpha$ is used for $D$ and $G$.

At step $t$, the discriminator is $\epsilon$-optimal

$$D(\boldsymbol{x}) - D(\boldsymbol{y}^{(t)}) = \boldsymbol{\theta}_D^\top \left( \boldsymbol{x} - \boldsymbol{y}^{(t)} \right) \geq \epsilon \tag{9}$$

$$\|\boldsymbol{\theta}_D\| \geq \frac{\epsilon}{\left\| \boldsymbol{x} - \boldsymbol{y}^{(t)} \right\|} \tag{10}$$

The gradients w.r.t. $\boldsymbol{\theta}_D$ and $\boldsymbol{\theta}_G$ are

$$\frac{\partial \mathcal{L}}{\partial \boldsymbol{\theta}_D} = \boldsymbol{x} - \boldsymbol{y}^{(t)} \tag{11}$$

$$\frac{\partial \mathcal{L}}{\partial \boldsymbol{\theta}_G} = \frac{\partial \mathcal{L}}{\partial \boldsymbol{y}^{(t)}} \times \boldsymbol{z}^\top$$

$$= \boldsymbol{\theta}_D \times \boldsymbol{z}^\top \tag{12}$$

If the learning rate $\alpha$ is small enough, $\left\| \boldsymbol{x} - \boldsymbol{y}^{(t)} \right\|$ should decrease as $t$ increases. As the empirical fake distribution converges to the empirical real distribution, $\left\| \boldsymbol{x} - \boldsymbol{y}^{(t)} \right\| \to 0$. The norm of gradient w.r.t. $\boldsymbol{\theta}_D$, therefore, decreases as $t$ increases and vanishes when the two empirical distributions are the same. From Eqn. 10, we see that, in order to maintain $D$'s $\epsilon$-optimality when $\left\| \boldsymbol{x} - \boldsymbol{y}^{(t)} \right\|$ decreases, $\|\boldsymbol{\theta}_D\|$ has to increase. From Eqn. 10 and 12, we see that the gradient w.r.t. $\boldsymbol{\theta}_G$ grows as the two empirical distributions are more similar. As $\left\| \boldsymbol{x} - \boldsymbol{y}^{(t)} \right\| \to 0$,

$$\frac{\left\| \frac{\partial \mathcal{L}_{\mathcal{W}}}{\partial \boldsymbol{\theta}_D} \right\|}{\left\| \frac{\partial \mathcal{L}_{\mathcal{W}}}{\partial \boldsymbol{\theta}_G} \right\|} \to 0 \tag{13}$$

Because the same learning rate $\alpha$ is used for both $G$ and $D$, $G$ will learn much faster than $D$. Furthermore, because $\left\| \boldsymbol{x} - \boldsymbol{y}^{(t)} \right\|$ decreases as $t$ increases, the difference

$$\frac{\epsilon}{\left\| \boldsymbol{x} - \boldsymbol{y}^{(t+1)} \right\|} - \frac{\epsilon}{\left\| \boldsymbol{x} - \boldsymbol{y}^{(t)} \right\|}$$

increases with $t$. The number of gradient steps that $D$ has to take to reach the next $\epsilon$-optimal state increases, and goes to infinity as $\left\| \boldsymbol{x} - \boldsymbol{y}^{(t)} \right\| \to 0$. Therefore, gradient descent with fixed number of updates to $\boldsymbol{\theta}_D$ cannot maintain the optimality of $D$.

The derivation for the objective in Eqn. 1 is similar.

## D  RESULTS ON DIFFERENT DATASETS

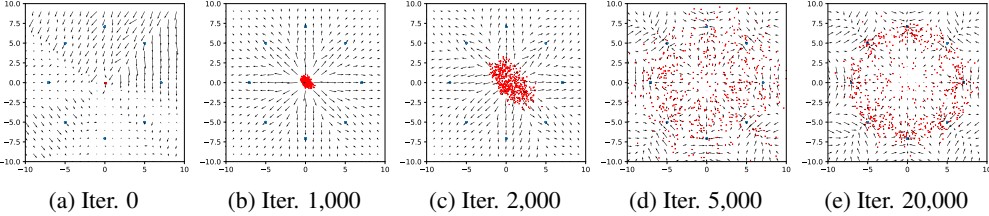

| (a) Iter. 0 | (b) Iter. 1,000 | (c) Iter. 2,000 | (d) Iter. 5,000 | (e) Iter. 20,000 |

Figure 5: Evolution of GAN-0-GP with $\lambda = 100$ on 8 Gaussians dataset.

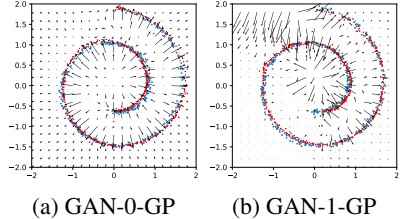

(a) GAN-0-GP          (b) GAN-1-GP

Figure 6: GANs trained with different gradient penalty on swissroll dataset. Although GAN-1-GP is able to learn the distribution, the gradient field has bad pattern. GAN-1-GP is more sensitive to change in hyper parameters and optimizers. GAN-1-GP fails to learn the scaled up version of the distribution.

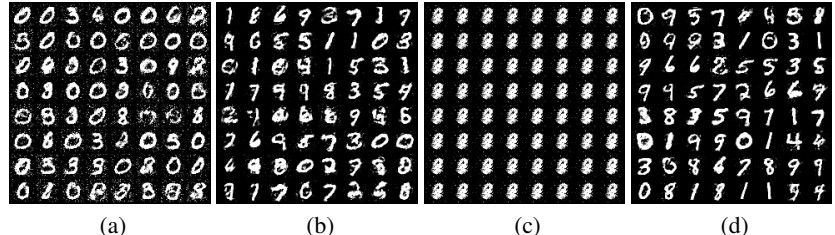

(a)              (b)              (c)              (d)

Figure 7: Result on MNIST. Adam was initialized with $\beta_1 = 0.5, \beta_2 = 0.9$. (a) No GP, iteration 10,000. (b) Zero-centered GP on real samples only with $\lambda = 100$, iteration 10,000. (c) One-centered GP with $\lambda = 100$, iteration 10,000. (d) Our zero-centered GP with $\lambda = 100$, iteration 10,000.

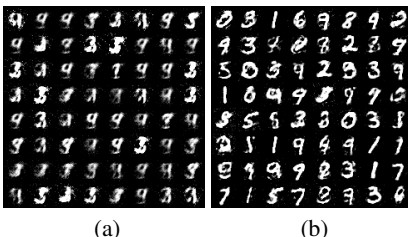

(a)              (b)

Figure 8: Result on MNIST. Adam was initialized with $\beta_1 = 0.9, \beta_2 = 0.99$. (a) Zero-centered GP on real samples only with $\lambda = 100$, iteration 10,000. (b) Our zero-centered GP with $\lambda = 100$, iteration 10,000.

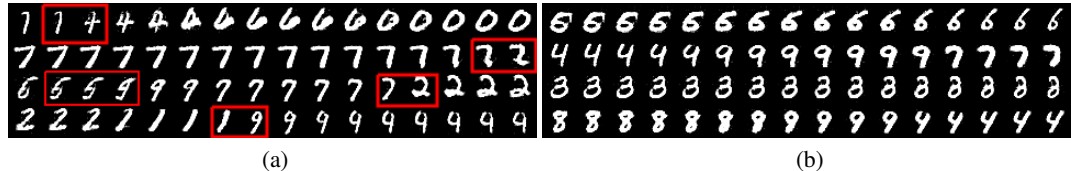

(a)                              (b)

Figure 9: Linear latent space interpolation between two random samples. (a) GAN with 0-GP-sample cannot perform smooth interpolation between modes. Small changes in input latent variable result in big difference in the output (red boxes). The result suggest that 0-GP-sample makes GANs to remember the training dataset and do not generalize to the region between samples in the training dataset. (b) GAN with our 0-GP can perform smooth interpolation between modes. The behavior implies that GANs with our 0-GP have better generalization.

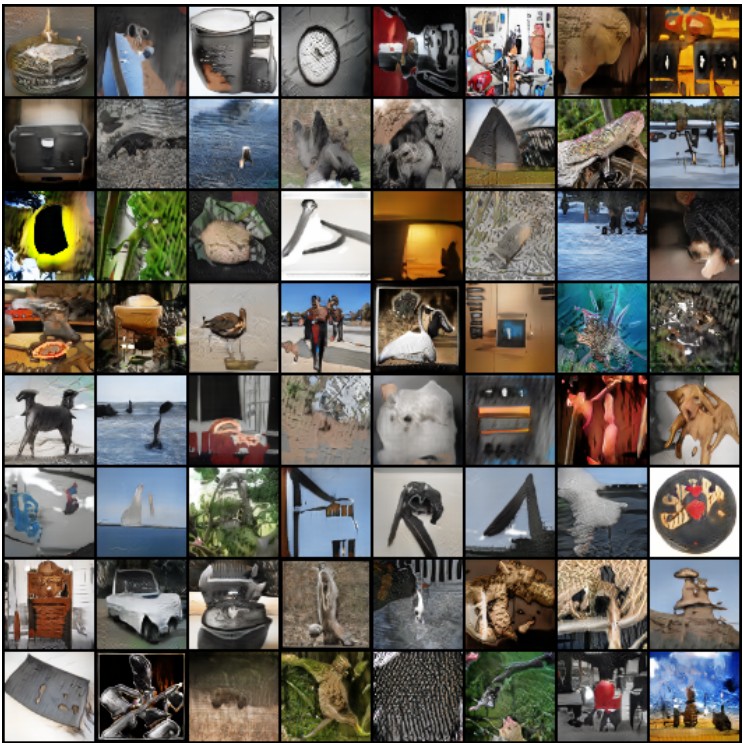

(a) All categories

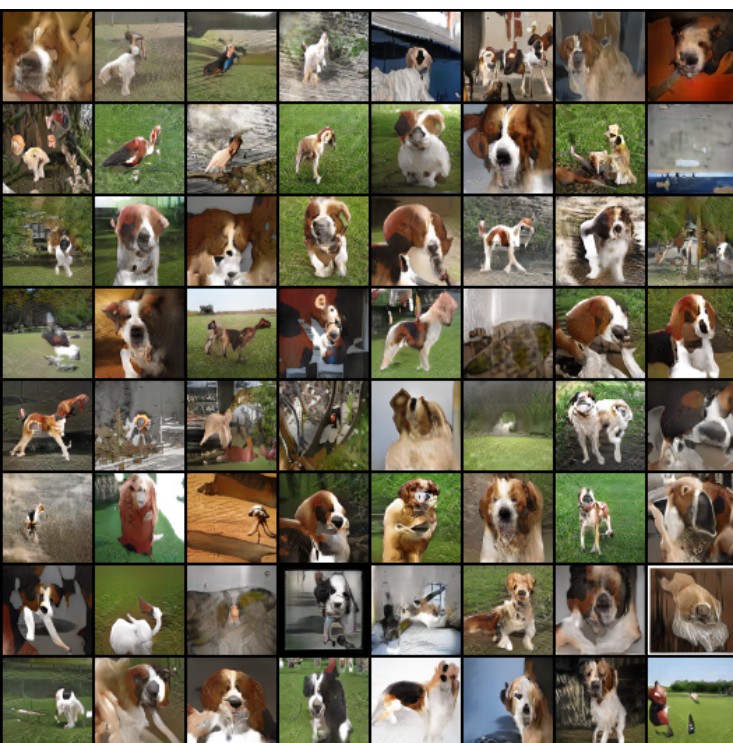

(b) Welsh springer spaniel

Figure 10: Samples from GAN-0-GP trained on ImageNet.

# E    IMPLEMENTATION DETAILS

We used Pytorch (Paszke et al., 2017) for development.

## E.1    SYNTHETIC AND MNIST DATASETS

Generator architecture in synthetic and MNIST experiments

| Fully connected layer $2 \times nhidden$ |
|---|
| $\downarrow$ |
| ReLU |
| $\downarrow$ |
| Fully connected layer $nhidden \times nhidden$ |
| $\downarrow$ |
| ReLU |
| $\downarrow$ |
| Fully connected layer $nhidden \times nhidden$ |
| $\downarrow$ |
| Fully connected layer $nhidden \times 2$ |

Discriminator architecture in synthetic and MNIST experiments

| Fully connected layer $2 \times nhidden$ |
|---|
| $\downarrow$ |
| ReLU |
| $\downarrow$ |
| Fully connected layer $nhidden \times nhidden$ |
| $\downarrow$ |
| ReLU |
| $\downarrow$ |
| Fully connected layer $nhidden \times nhidden$ |
| $\downarrow$ |
| Fully connected layer $nhidden \times 1$ |
| $\downarrow$ |
| Sigmoid |

Hyper parameters for synthetic and MNIST experiments

| Learning rate | 0.003 for both $G$ and $D$ |
|---|---|
| Learning rate TTUR | 0.003 for $G$, 0.009 for $D$ |

## E.2    IMAGENET

The entire ImageNet dataset with all 1000 classes was used in the experiment. Because of our hardware limits, we used images of size $64 \times 64$. We used the code from Mescheder et al. (2018), available at `https://github.com/LMescheder/GAN_stability`, for our experiment. Generator and Discriminator are ResNets, each contains 5 residual blocks. All GANs in our experiment have the same architectures and hyper parameters. The configuration for WGAN-GP5 is as follows.

```
generator:
  name: resnet2
  kwargs:
    nfilter: 32
    nfilter_max: 512
    embed_size: 128
discriminator:
  name: resnet2
  kwargs:
```

```
    nfilter: 32
    nfilter_max: 512
    embed_size: 128
z_dist:
  type: gauss
  dim: 128
training:
  out_dir: ../output/imagenet_wgangp5_TTUR
  gan_type: wgan
  reg_type: wgangp
  reg_param: 10.
  batch_size: 64
  nworkers: 32
  take_model_average: true
  model_average_beta: 0.999
  model_average_reinit: false
  monitoring: tensorboard
  sample_every: 1000
  sample_nlabels: 20
  inception_every: 10000
  save_every: 900
  backup_every: 100000
  restart_every: -1
  optimizer: adam
  lr_g: 0.0001
  lr_d: 0.0003
  lr_anneal: 1.
  lr_anneal_every: 150000
  d_steps: 5
  equalize_lr: false
```

## F  FINDING A BETTER PATH BETWEEN A PAIR OF SAMPLES

Because the set of real data is unlikely to be convex, a linear interpolation between two datapoints is unlikely to be in the set. For example, the weighted average of two real images is often not a real image. If the generator is good enough then its output is likely to be on the real data manifold. A path $\mathcal{C}$ from $\boldsymbol{y}$ to $\boldsymbol{x}$ can be found by transforming the straight line between the two corresponding latent codes, to the data space using the generator $G$. To get the latent code of real datapoints, we use an encoder $E$ which is trained to map real data to normally distributed latent codes. The pseudo code of the path finding algorithm is shown in Alg. 1.

**Data:** an encoder $E$; a pair of samples $\boldsymbol{x}, \boldsymbol{y} = G(\boldsymbol{z})$;
**Result:** interpolated datapoint $\tilde{\boldsymbol{x}}$
1 Get the latent code of $\boldsymbol{x}$: $\boldsymbol{z}_x = E(\boldsymbol{x})$
2 Calculate the interpolated latent code: $\tilde{\boldsymbol{z}} = \alpha\boldsymbol{z}_x + (1 - \alpha)\boldsymbol{z}$
3 Generate the interpolated datapoint: $\tilde{\boldsymbol{x}} = G(\tilde{\boldsymbol{z}})$
**Algorithm 1:** Path finding algorithm

