# OpenReview forum: "Improving Generalization and Stability of Generative Adversarial Networks"
_ICLR.cc/2019/Conference_

### Official Review · AnonReviewer3 · 2018-10-26
**An interesting read on the convergence of GANs with gradient penalties, lacking comparisons to WGAN-GP**

**Rating:** 6
**Confidence:** 4

**Review:**

Summary:
The paper proposes to add to the original GAN (2014) loss a zero-centered gradient penalty as the one defined in the WGAN-GP paper. It also provides an analysis on the mode collapse and lack of stability of classical GANs. The authors compare results using their penalty on a few synthetic examples and on image net dogs generations to results using the classical GAN loss with or without gradient penalties.

Positive points:
The paper is interesting to read and well illustrated.
An experiment on imagenet illustrates the progress that can be achieved by the proposed penalty.

Points to improve:

If I understood correctly, the main contribution resides in the application of the GP proposed by WGAN-GP to the original setting. Why not compare results to WGAN-GP in this case? Since the proposal of GANs, many papers addressed the mode collapse problem. WGAN-GP, VEEGAN, or Lucas et al arXiv:1806.07185, ICML 2018 to name only a few.
The related work section looks incomplete with some missing related references as mentioned above, and copy of a segment that appears in the introduction.
The submission could maybe improved by segmenting the work into intro / related / background (with clear equations presenting the existing GP) / analysis / approach / experiments
The experiments on synthetic data could be improved: for reproducibility, many works on GANs used the same synthetic data as VEEGAN.
The imagenet experiment lacks details.

---

> ### Author Response · Authors · 2018-11-07
> **Response to reviewer 3**
>
> Thank you for your comments. We would like to address your concerns as follow.
>
> 1. We do not use the gradient penalty in WGAN-GP (1-GP) to improve the original GAN. Our 0-GP, although has a similar form as 1-GP,  is motivated from a very different perspective and produces very different effects. We assume that you find our 0-GP similar to 1-GP because of the use of the straight line from a fake to a real sample. In the response to reviewer 1, we propose a more sophisticated way to find a path from a fake to a real datapoint. The new method highlights the difference between our method and 1-GP.
>
> 2. The 0-GP is not the only contribution of our paper. We start by analyzing the generalization of GANs, showing the problem of the original GAN loss. Although generalizability is one of the most desirable properties of generative models, it has not been studied carefully in GAN literature. Based on our analysis, we propose 0-GP to improve the generalization of GANs. On the 8 Gaussian dataset, GAN-0-GP can generate plausible unseen datapoints on the circle, implying better generalization. We show that the original GAN loss makes GAN focuses on generating datapoints in the training dataset. 0-GP-sample proposed in [4] encourages the generator to remember the training samples. That result in the mode jumping behavior: when we perform interpolation between $z_1$ and $z_2$, the output does not smoothly transform from $x_1 = G(z_1)$ to $x_2 = G(z_2)$ but suddenly jump from $x_1$ to $x_2$. The behavior can be seen in figure 8 of BigGAN paper (https://arxiv.org/abs/1809.11096).
>
> 3. We will include WGAN-GP to the baselines for the sake of completeness. However, as discussed in the previous paragraphs and in our paper, WGAN-GP and their 1-GP does not address the same problem as our 0-GP.
>
> As discussed in our paper, 1-GP does not help improving generalization in GANs. [4] even showed that 1-GP does not help WGAN (and the original GAN as well) to converge to an equilibrium. The phenomenon can be seen in our MNIST experiment where GAN-1-GP fails to produce any realistic samples after 10,000 iterations. It has been observed that WGAN-1-GP does not converge to an equilibrium, the generator continues to map the same noise to different modes as the training continues. In our synthetic experiment, WGAN-1-GP is less robust to change in hyper-parameters than GAN-0-GP. Detailed results will be included in our revision. Please refer to [4] for more in-depth discussion about the non-convergence of WGAN-GP.
>
> When $p_g$ is the same as $p_r$, the gradient of the optimal discriminator in GAN and the optimal critic in WGAN must be 0. Any non-zero centered GP will not help GANs to converge to the optimal equilibrium.  Our 0-GP helps to improve both generalization and convergence of GANs. Our 0-GP can be applied to WGAN as well.
>
> Similar to the original GAN, WGAN and WGAN-GP can overfit to the dataset: the distance output by the critic can be larger than the Wasserstein distance between the two distributions. However, overfitting in WGAN and WGAN-GP is not as severe as in GAN. This is partly because the gradient in WGAN and WGAN-GP does not explode so mode collapse is much harder to observe.
>
> 4. We will include more related works to our paper. The vast body of work on GANs makes it difficult to find all related works. We only focus on some key papers on the topic.
>
> Discussion about VEEGAN and Lucas et al. will be added to our next revision. However, we want to emphasize that our work is about improving the generalization of GANs. Reducing mode collapse is related to but is not exactly the same as generalization. As in the 8 Gaussian dataset, a GAN without mode collapse is the one that can generate all 8 modes. A GAN with good generalization should be able to generate unseen datapoints on the circle and to perform smooth interpolation between modes.
>
> 5. We will add more details about the experiments to the appendix. The code for all experiments will be released after the review process. For the imagenet experiment, we used the code from [4] which is available on github. We note that [4] is a state-of-the-art method which is able to help GAN to scale to massive datasets and it is used in BigGAN paper.
>
> 6.Thank you for your suggestion about the paper layout. Adding a table that summarizes referred gradient penalties is a good idea.

---

> ### Author Response · Authors · 2018-11-19
> **Revision uploaded**
>
> Thank you again for your suggestions. We have revised our paper to address your concerns as follows:
>
> 1. A background section is added with basic information about GANs and a definition of generalization. A table summarizing the referred gradient penalties is also added.
>
> 2. We extended the Related works section to include papers which address the mode collapse problem. The writing of this part and the whole paper was revised.
>
> 3. Another MNIST experiment is added to Section 6.1 to further demonstrate the effectiveness of our method in preventing overfitting. Specifically, our GAN-0-GP is the only GAN that could learn to generate realistic samples when the discriminator is updated 5 times per generator update.
>
> 4. WGAN-GP is included to our ImageNet experiment. Our GAN-0-GP outperforms WGAN-GP by a large margin.
>
> 5. Implementation details are added to the appendix. The code for all experiments will be released after the review process.
>
> 6. We added the analysis for the 'mode jumping' problem to Section 6.2. We showed that GAN-0-GP-sample suffers from the problem. On the other hand, our GAN-0-GP is robust to the problem and is able to produce better interpolation between modes.
>
> 7. A new algorithm for finding a better path between a pair of samples is added to our paper.

---

> > ### Comment · AnonReviewer3 · 2018-11-26
> > **Thank you for the revision, a key comparison is still missing.**
> >
> > I appreciate the authors effort put in the revised pdf.
> >
> > I acknowledge the comparison with WGAN-GP in the imageNet experiment.
> > The point that still bothers me in the main results of the paper (generalization) is the lack of comparison with WGAN-GP in Figure 1: the comparison with 1GP is displayed, but this is with a classical GAN objective, and not a WGAN loss. My insight is that the results should be better using the full WGAN-GP objective. Sorry for the last minute request, but this doesn't seem like a difficult comparison to make.
> >
> > Minor but still important to reader understanding:
> > - Table 1 is nice but is not referenced in the text. The caption is not very explicit either. A ref should be added for the 0-GP sample (name only introduced in section 5). For the 1-GP a ref to WGAN-GP could be added too with precisions on the full criterion in the caption.
> > - Probably will lack of time, but an interesting experiments for assessing GAN generalization could be to measure the Mean reconstruction error as in
> > Guo-Jun Qi. Loss-Sensitive Generative Adversarial Networks, 2017.
> >
> > Other comments:
> > p1: typos equilibira
> > GAN loss tends -> GAN loss tend
> > p10: add the dataset in caption of Figure 4
> > Remove urls in the bib

---

> > > ### Author Response · Authors · 2018-11-26
> > > **Response to reviewer 3**
> > >
> > > Thank you for pointing out the typos. We have fixed them in our latest revision. We also added a reference to table 1 in our introduction.
> > > We would like to address your other concerns as follows:
> > >
> > > 1.  Figure 1 aims to illustrate the effect of our 0-GP in pushing the discriminator toward the theoretically optimal discriminator $D*$ in the original GAN, not the theoretically optimal critic $C*$ in WGAN.
> > > If we add WGAN-GP to the comparison in Fig. 1, it should be compared with the optimal critic, not the optimal discriminator in Fig. 1f. However, we are not aware of any closed-form formula for the Wasserstein-1 distance between two multivariate Gaussians. Therefore, WGAN-GP could not be added to Fig. 1. Also note that the critic can output values outside of the [0, 1] range so it is not comparable to discriminators in Fig. 1.
> > >
> > > We also do not see the benefit of adding WGAN-GP to the comparison in Fig. 1. Our work focuses on the use of GP in improving generalization in GAN, not on the use of metric/divergence. As shown in section 5.3, 1-GP does not help to prevent overfitting in the discriminator with the original GAN loss. Similar to the discriminator, the critic in WGAN-GP can also overfit to the data and output a distance greater than the actual Wasserstein-1 (W-1) distance between $p_g$ and $p_r$. For example, $p_g$ and $p_r$ are both standard normal distribution, the W-1 distance between them is 0. However, the critic is trained to maximize the distance between the two empirical distributions can output a value greater than 0 without violating the Lipschitz constraint. More concretely, if the dataset contains a real sample $x$ and a fake sample $y$, the empirically optimal critic satisfies:
> > >                                    |C(x) - C(y)| = || x - y ||,
> > > while the true distance is 0. The phenomenon makes the critic and the generator to oscillate around the equilibrium [4] and slows down the improvement in sample quality. Note that, WGAN-LP [5] with the corrected GP still suffers from that same problem and performs similarly to WGAN-GP on real datasets. The problem is seen in our ImageNet experiment where GAN-0-GP and GAN-0-GP-sample outperform WGAN-GP by a large margin.
> > >
> > > 2. We appreciate your suggestion on improving the paper quality. We will add more experiments if time permits. However, we believe that the current experiments are sufficient to demonstrate our main contribution which is a novel method for improving the generalization of GANs.
> > >
> > > [4] Which Training Methods for GANs do actually Converge?
> > > Lars Mescheder, Andreas Geiger, Sebastian Nowozin.
> > > [5] On the regularization of Wasserstein GANs
> > > Henning Petzka, Asja Fischer, Denis Lukovnicov.

---

> > > > ### Comment · AnonReviewer3 · 2018-11-28
> > > > **Increased score in the light of the comparison with WGAN-GP**
> > > >
> > > > I get the authors argument about the optimal results that couldn't be shown in Fig.1 for the Wasserstein 1-GP loss, but I still think this comparison could be interesting, and encourage the authors to add it to the camera ready version if the paper is accepted.
> > > > Nonetheless, since the authors included a comparison to Wasserstein-GP on ImageNet, I increased my rating.

---

> > > > > ### Author Response · Authors · 2018-12-03
> > > > > **Comparison to WGAN-GP**
> > > > >
> > > > > Thank you for increasing the rating. We have performed another experiment to compare the generalization of GANs. The detail of the experiment is as follows:
> > > > >
> > > > > 1. Experiment setup
> > > > > Dataset: We stack 3 MNIST images into an RGB image, resulting in a dataset of 1000 major modes.
> > > > > Network architecture: Generator and Discriminator/Critic are 2 hidden layer MLPs with 512 hidden neurons.
> > > > > A CNN classifier classifies each channel of the generated image into 1 of the 10 classes. The test set accuracy of the classifier is 99%. The classifier allows us to automatically count the number of modes in the model distribution.
> > > > > To evaluate a generator, we generate 100,000 samples and count the number of modes in that generated dataset. On average, each mode contains 100 samples. To counter the inaccuracies in the classifier, a mode is counted as present if there are more than 10 samples fall into that mode.
> > > > >
> > > > > 2. Result
> > > > > Results were averaged between 3 runs.
> > > > >
> > > > > The number of modes after 10,000 generator iterations:
> > > > > GAN-0-GP with lambda = 500, 1 discriminator iterations per generator iteration: 943
> > > > > GAN-0-GP with lambda = 500, 5 discriminator iterations per generator iteration: 1000
> > > > > GAN-0-GP with lambda = 50, 5 discriminator iterations per generator iteration: 1000
> > > > > GAN-0-GP with lambda = 10, 5 discriminator iterations per generator iteration: 984
> > > > > WGAN-0-GP with lambda = 500, 5 critic iterations per generator iteration: 1000
> > > > > WGAN-0-GP with lambda = 50, 5 critic iterations per generator iteration: 1000
> > > > > WGAN-0-GP with lambda = 10, 5 critic iterations per generator iteration: 1000
> > > > > WGAN-1-GP with lambda = 500, 5 critic iterations per generator iteration: 847
> > > > > WGAN-1-GP with lambda = 50, 5 critic iterations per generator iteration: 996
> > > > > WGAN-1-GP with lambda = 10, 5 critic iterations per generator iteration: 1000
> > > > >
> > > > > 0-GP improves the performance of the original GAN as well as WGAN.  The penalty weight lambda in WGAN-1-GP (WGAN-GP) is hard to tune. Larger values of lambda make WGAN-1-GP to oscillate. WGAN-0-GP performance is consistent across various settings. It also reaches 1000 modes sooner than WGAN-1-GP.
> > > > >
> > > > > Increasing the number of discriminator iterations per generator iteration actually improve the performance of GAN-0-GP. We would like to recall that increasing the number of discriminator iterations hurts the performance of the original GAN and GAN-0-GP-sample as it makes the discriminator overfit to the training samples.
> > > > >
> > > > > From the result, we can conclude that 0-GP helps improving generalization of GAN.

---

### Official Review · AnonReviewer1 · 2018-11-01
**good discussion of generalization and stability of GAN and the gradient penalty method is promising**

**Rating:** 7
**Confidence:** 3

**Review:**

The paper discusses the generalization capability of GAN especially from the discriminator's perspective. The explanation is clear and the method is promising. The proposed gradient penalty method that penalizes the unseen samples is novel and reasonable from the explanation, although these methods has been proposed before in different forms.

Pros:
1. Nice explanation of why the training of GAN is not stable and the modes often collapse.
2. Experiments show that the new 0-gradient penalty method seems promising to improve the generalization capability of GAN and helps to resist mode collapsing.

Cons:
1. The paper does not have a clear definition of the generalization capability of the network.
2. The straight line segment between real and fake images seems not a good option as the input images may live on low-dimensional manifolds.
3. Why samples alpha in (7) uniformly? It seems the sampling rate should relate with its value. Intuitively, the closer to the real image the sampling point is, the larger the penalty should be.

---

> ### Author Response · Authors · 2018-11-07
> **Response to reviewer 1**
>
> Thank you for your constructive comments. We would like to address your concerns as follow:
> 1. Generalization has been defined in [1, 2, 3]. They were cited in our paper. Because of the space limit, we could not include their definition in the first version of our paper. We will add the definition from [1] to the updated version. The definition in [1] is directly related to our discussion: if the Lipschitz constant is 0, then the network has the maximum generalization capability and no discriminative power. As stated in our paper, our gradient penalty makes the network generalizable while remaining discriminative.
>
> 2. We agree that the straight line is not a good option for real data like images. However, it's the cheapest way to implement our method. We are working on an improved version for the GP, which we briefly describe below. We plan to include the result in the next revision of our paper.
>
> For all interpolated points to be in the same set with the two endpoints, the set must be convex. $supp(p_g) U supp(p_r)$ is generally not convex so linear interpolation in the data space cannot guarantee that every interpolated point to be in the support.  A solution to this problem is to force the set of latent code $z$ to be convex and perform the interpolation in the latent space. This requires an additional encoder E to encode a datapoint $x$ to a latent code $z_x$. The process of sampling a datapoint for regularization is as follow
> (i) Sample a noise vector $z ~ p_z$, generate a fake datapoint $y = G(z)$
> (ii) Sample a real datapoint $x ~ p_r$, get the latent code of $z_x = E(x)$
> (iii) Generate the interpolated latent code: $\tilde(z) = \alpha z_x + (1 - \alpha) z$
> (iv) Generate the interpolated datapoint: $\tilde(x) = G(\tilde(z))$
> (v) Apply gradient penalty on $\tilde(x)$
> $\tilde(x)$ is more likely to lie on the data manifold than the weighted sum of a real and a fake sample. Regularizing the gradient w.r.t. $\tilde(x)$ will allow better generalization and discrimination.
>
> 3. We are not sure that your suggestion is correct. As discussed in our paper, gradient exploding tends to happen near the decision boundary, while the gradient near real/fake datapoints tends to vanish. We doubt that increasing the sampling rate near real/fake datapoints will lead to better result.
>
>
> [1] Generalization and Equilibrium in Generative Adversarial Nets (GANs). Sanjeev Arora, Rong Ge, Yingyu Liang, Tengyu Ma, Yi Zhang.
> [2] Do GANs actually learn the distribution? An empirical study. Sanjeev Arora, Yi Zhang.
> [3] On the Discrimination-Generalization Tradeoff in GANs. Pengchuan Zhang, Qiang Liu, Dengyong Zhou, Tao Xu, Xiaodong He.
> [4] Which Training Methods for GANs do actually Converge? Lars Mescheder, Andreas Geiger, Sebastian Nowozin.

---

> ### Author Response · Authors · 2018-11-19
> **Revision uploaded**
>
> Thank you for your review and questions. We have performed additional experiments and analysis to consolidate our finding. The updates are as follows:
>
> 1. A background section is added with basic information about GANs and a definition of generalization. A table summarizing the referred gradient penalties is also added.
>
> 2. We extended the Related works section to include papers which address the mode collapse problem. The writing of this part and the whole paper was revised.
>
> 3. Another MNIST experiment is added to Section 6.1 to further demonstrate the effectiveness of our method in preventing overfitting. Specifically, our GAN-0-GP is the only GAN that could learn to generate realistic samples when the discriminator is updated 5 times per generator update.
>
> 4. WGAN-GP is included to our ImageNet experiment. Our GAN-0-GP outperforms WGAN-GP by a large margin.
>
> 5. Implementation details are added to the appendix. The code for all experiments will be released after the review process.
>
> 6. We added the analysis for the 'mode jumping' problem to Section 6.2. We showed that GAN-0-GP-sample suffers from the problem. On the other hand, our GAN-0-GP is robust to the problem and is able to produce better interpolation between modes.
>
> 7. A new algorithm for finding a better path between a pair of samples is added to our paper.

---

### Official Review · AnonReviewer2 · 2018-11-02
**Solid analytical and experimental exploration concerning GAN generalizability**

**Rating:** 7
**Confidence:** 3

**Review:**

The primary innovation of this paper seems focused towards increasing the generalization of GANs, while also maintaining convergence and preventing mode collapse.

The authors first discuss common pitfalls concerning the generalization capability of discriminators, providing analytical underpinnings for their later experimental results. Specifically, they address the problem of gradient explosion in discriminators.

The authors then suggest that a zero-centered gradient penalty (0-GP) can be helpful in addressing this issue. 0-GPs are regularly used in GANs, but the authors point out that the purpose is usually to  provide convergence, not to increase generalizability. Non-zero centered penalties can give a convergence guarantee but, the authors, assert, can allow overfitting. A 0-GP can give the same guarantees but without allowing overfitting to occur.


The authors then verify these assertions through experimentation on synthetic data, as well as MNIST and ImageNet. My only issue here is that very little information was given about the size of the training sets. Did they use all the samples? Some portion? It is not clear from reading. This would be a serious impediment to reproducibility.

All in all, however, the authors provide a convincing  combination of analysis and experimentation. I believe this paper should be accepted into ICLR.

Note: there is an error on page 9, in Figure 3. The paragraph explanation should list that the authors' 0-GP is figure 3(e). They list (d) twice.

---

> ### Author Response · Authors · 2018-11-07
> **Response to reviewer 2**
>
> Thank you for your review. We will revise our paper according to your suggestion. We would like to quickly address your question about the experiment here. For MNIST and ImageNet experiment, the whole dataset was used. For the ImageNet experiment, we used the code from [4]. Details about all experiments will be added to the appendix. We thank you for pointing the typo in Figure 3.
>
> We will also add an in-depth discussion about our method and other related works to our next revision as suggested by other reviewers.

---

> ### Author Response · Authors · 2018-11-19
> **Revision uploaded**
>
> Thank you for your constructive review. We have updated our paper to address your concerns. The changes are summarized as follow:
>
> 1. A background section is added with basic information about GANs and a definition of generalization. A table summarizing the referred gradient penalties is also added.
>
> 2. We extended the Related works section to include papers which address the mode collapse problem. The writing of this part and the whole paper was revised.
>
> 3. Another MNIST experiment is added to Section 6.1 to further demonstrate the effectiveness of our method in preventing overfitting. Specifically, our GAN-0-GP is the only GAN that could learn to generate realistic samples when the discriminator is updated 5 times per generator update.
>
> 4. WGAN-GP is included to our ImageNet experiment. Our GAN-0-GP outperforms WGAN-GP by a large margin.
>
> 5. Implementation details are added to the appendix. The code for all experiments will be released after the review process.
>
> 6. We added the analysis for the 'mode jumping' problem to Section 6.2. We showed that GAN-0-GP-sample suffers from the problem. On the other hand, our GAN-0-GP is robust to the problem and is able to produce better interpolation between modes.
>
> 7. A new algorithm for finding a better path between a pair of samples is added to our paper.

---

### Public Comment · ~Seoungyoon_Kang1 · 2018-11-27
**Question about implementing detail: How do you train encoder E to get latent vector of real image X?**

Hi, I read your paper and it think it is quiet well written paper.
I would like to ask 2 questions about your paper.

1. Additional information about your generating performance
In the PDF of your paper in OpenReview, it is hard to compare generation results with other papers.
It would be better if the paper contains large resolution results with which resolution it is.

2. About encoder E
In the paper, to penalize gradients on points in line segment between Y(t) and X, you proposed linear interpolation between two real and fake latent code. In appendix F, you explained how you get latent code ${z}_{x} using encoder E, however, details of E is not in the paper. How did you train E? It doesn't seem to be pretrained. Is it able to infer ${z}_{x} correctly? If so, how did you schedule the training step? Separate D and E or train at once?

Thanks in advance.

---

> ### Author Response · Authors · 2018-11-28
> **Training encoder E**
>
> Thank you for your comment. The encoder E could be trained jointly with D and G using an architecture similar to BiGAN [6].
>
> [6] Adversarial Feature Learning
> Jeff Donahue, Philipp Krähenbühl, Trevor Darrell

---

### Meta-Review · Area_Chair1 · 2018-12-16
**Unanimous accept.**

**Confidence:** 4
**Recommendation:** Accept (Poster)

**Metareview:**

The paper received unanimous accept over reviewers (7,7,6), hence proposed as definite accept.